# Quantitative MRI of a Cerebral Cryptococcoma Mouse Model for In Vivo Distinction between Different Cryptococcal Molecular Types

**DOI:** 10.3390/jof10080593

**Published:** 2024-08-22

**Authors:** Luigi Musetta, Shannon Helsper, Lara Roosen, Dries Maes, Anca Croitor Sava, Liesbeth Vanherp, Willy Gsell, Greetje Vande Velde, Katrien Lagrou, Wieland Meyer, Uwe Himmelreich

**Affiliations:** 1Biomedical MRI, Department of Imaging and Pathology, KU Leuven, 3000 Leuven, Belgium; luigi.musetta@kuleuven.be (L.M.); shannon.helsper@kuleuven.be (S.H.); lara.roosen@kuleuven.be (L.R.); dries.maes@kuleuven.be (D.M.); anca.croitor@kuleuven.be (A.C.S.); liesbeth.vanherp@kuleuven.be (L.V.); willy.gsell@kuleuven.be (W.G.); greetje.vandevelde@kuleuven.be (G.V.V.); 2µNEURO Research Centre of Excellence, University of Antwerp, 2000 Antwerp, Belgium; 3Laboratory of Clinical Microbiology, Department of Microbiology, Immunology and Transplantation, KU Leuven, 3000 Leuven, Belgium; katrien.lagrou@uzleuven.be; 4Department of Laboratory Medicine, National Reference Center for Mycosis, UZ Leuven, 3000 Leuven, Belgium; 5Westerdjjk Fungal Biodiversity Institute—KNAW, 3584 CT Utrecht, The Netherlands; w.meyer@wi.knaw.nl

**Keywords:** cryptococcosis, murine models, magnetic resonance imaging

## Abstract

The controversially discussed taxonomy of the *Cryptococcus neoformans*/*Cryptococcus gattii* species complex encompasses at least eight major molecular types. Cerebral cryptococcomas are a common manifestation of cryptococcal neurological disease. In this study, we compared neurotypical symptoms and differential neurovirulence induced by one representative isolate for each of the eight molecular types studied. We compared single focal lesions caused by the different isolates and evaluated the potential relationships between the fungal burden and properties obtained with quantitative magnetic resonance imaging (qMRI) techniques such as diffusion MRI, T_2_ relaxometry and magnetic resonance spectroscopy (MRS). We observed an inverse correlation between parametric data and lesion density, and we were able to monitor longitudinally biophysical properties of cryptococcomas induced by different molecular types. Because the MRI/MRS techniques are also clinically available, the same approach could be used to assess image-based biophysical properties that correlate with fungal cell density in lesions in patients to determine personalized treatments.

## 1. Introduction

Isolates of the *Cryptococcus neoformans*/*Cryptococcus gattii* species complexes (CN/CG) are opportunistic pathogenic yeasts found in the environment (e.g., soil, decaying wood, bird guano). The infectious particles present in the environment enter the host initially through inhalation. They can cause lung infections and disseminate with a remarkable tropism for the central nervous system (CNS) [1]. Cryptococcal meningoencephalitis is globally the most common cause of CNS fungal infection, which mainly affects immunocompromised people such as HIV-infected individuals and solid organ transplant recipients [2,3]. Once in the brain, the opportunistic yeast can lead to the development of meningoencephalitis and/or induce a typical spherical lesion in the brain parenchyma called cryptococcomas. The latter typically consists of a local accumulation of cryptococci encapsulated in a lytic globular structure composed of gelatinous mucoid material and occasionally inflammatory cells [4,5].

In 2022, the WHO (World Health Organization) published, for the first time since its foundation, a fungal priority list, ranking CN and CG in the critical and medium priority groups, respectively. In this report, there is a clear request to better understand the antifungal susceptibility of different molecular types [6,7]. The complex relationship of different isolates has resulted in repeated proposals for revisions of the taxonomy of the CN/CG species complexes and has been part of a continuous debate [8,9,10,11,12,13,14,15,16]. The consensus on multi-locus sequence typing highlighted a differential geographical distribution as well as different associations with the clinical manifestation of symptoms [17,18,19,20].

While CN and CG result in partially different symptoms and manifestations of cryptococcosis in patients [10,11], these have not been studied systematically for the major different molecular types. In addition, confounding factors like underlying diseases, geographic location and others make it difficult to assess the pathogenesis of different molecular types of CN/CG isolates. Preclinical studies in rodent models provide the opportunity to assess differences between the major molecular types in the pathogenesis of cryptococcosis in vivo under controlled conditions [21,22,23,24,25]. However, due to differences in methodology, mouse models, and the use of only very few cryptococcal isolates, a systematic comparison to evaluate differences among the eight major molecular types in vivo using animal models is lacking. Limitations of murine models can include the use of invasive methods that are restricted to single time points relying on survival curves to define virulence, which exclude fundamental information to delineate the state of progression of the disease and dissemination [26,27,28,29]. Pool et al. provided criteria to evaluate the neurotypical symptoms of cryptococcosis between genetically modified strains, with the potential to also discriminate differences in neurovirulence with disease progression across the different major molecular types of the cryptococcus species complexes [30].

Recently, our group investigated in vivo imaging techniques to assess the pathogenesis, dissemination, and treatment of cryptococcosis in murine models non-invasively and longitudinally [31,32,33,34,35,36]. Hereby, quantitative magnetic resonance imaging (qMRI) methods are of particular interest in assessing cerebral cryptococcosis due to its direct translatability to human patients. qMRI refers to non-invasive MRI techniques that allow quantitative measurements of the biophysical properties of tissues under in vivo conditions both in animal models and in the clinic. qMRI holds the remarkable potential to discriminate between different tissue types, as well as different lesions, based on their imaging properties, either complementing or replacing biopsies [5,37]. qMRI techniques, including diffusion MRI, MR relaxometry, and spectroscopy, have been used to assess cell density or related capsule size within fungal lesions of a living host, enabling repeated and hereby longitudinal assessment of physical and metabolic changes of the lesion content in vivo [34,38]. These properties correlate with the concentration of fungal metabolites like trehalose or mannitol, as well as the colony-forming units (CFUs) in cryptococcomas [5,34,39,40,41].

While differences between reference strains of CN and CG, namely H99 and R265, are frequently studied, no systematic evaluation of virulence, pathogenesis, and disease progression among the different major molecular types of the CN/CG species complexes has been performed. In this study, we characterized differences in pathogenesis, lesion formation and biophysical properties of cryptococcomas in the mouse brain in vivo using qMRI techniques and investigated potential differences between cerebral cryptococcomas caused by the eight molecular types of CN/CG (VN I-IV; VG I-IV). We investigated the evolution of fungal lesion formation over time and, in addition, investigated non-invasively how MR properties potentially characterize the major molecular types of the CN/CG species complexes. We correlated brain fungal burden and the lesion fungal density with the clinically available qMRI techniques, delineating unique lesion biophysical properties.

## 2. Materials and Methods

### 2.1. Fungal Culture and Inoculum Preparation

Cryptococcal isolates, each representing one of the eight major molecular types, were received from the Molecular Medical Mycology Research Laboratory at Westmead Institute for Medical Research, University of Sydney, Australia [17] (Table 1). The stock cultures were stored at −80 °C suspended in a 10% (*w*/*v*) milk powder solution in sterile phosphate-buffer saline (PBS; 1X, GIBCO^®^, Life Technologies-Invitrogen, Paisley, UK). After thawing, the isolates were plated on Sabouraud agar (2% agar, following manufacturer instructions, product number: 64644, Biorad, Hercules, CA, USA) and incubated upside-down at 30 °C for 2 to 3 days. After the incubation period, colonies were transferred and resuspended into a liquid Sabouraud growth medium (2% glucose, following manufacturer instructions, product number CM0147, Oxoid, UK). The samples were vortexed and incubated for another 2 days at 30 °C. After the incubation period, samples were centrifuged (5 min; 650× *g*; 20 °C), and the formed pellet was washed twice with sterile PBS and again centrifuged (5 min; 650× *g*; 20 °C). The final pellet was resuspended in 10 mL PBS, and 10 µL of the fungal suspension was counted using a Neubauer-improved chamber (Marienfeld, Lauda-Königshofen, Germany). The suspension was then further diluted to obtain the desired inoculum of 100 µL volume containing 10^4^ cells/µL in sterile PBS. The viability of the cryptococci in the inoculum was confirmed by the number of CFUs in a 10-fold dilution series starting with 20 µL of the inoculum plated on Sabouraud agar. Colonies were counted manually after 3 to 4 days of incubation upside-down at room temperature.

### 2.2. Stereotactic Injection in Mice

Animal experiments were performed in accordance with the national regulations and the European Directive 2010/63/EU [42] and after approval by the Animal Ethics Committee of KU Leuven under project number p197/2021. Female BALB/cAnNCrl mice (n = 4 per isolate, 32 mice total) at the age of 8 to 10 weeks (internal stock KU Leuven, Leuven, Belgium; Charles River Laboratories) were used for all experiments. The mice were housed in ventilated cages with food and water ad libitum and a maximum of four animals per cage. For surgical procedures, the animals were first anesthetized using an intraperitoneal injection of ketamine (40 to 60 mg/kg; Nimatek, Eurovet Animal Health, Bladel, The Netherlands) and medetomidine (0.05 to 0.10 mg/kg; Domitor, Orion Pharma, Espoo, Finland). The animals were then fixed on a stereotactic head frame (Stoelting, Wood Dale, IL, USA). A dosage of 30 µL Lidocaine (Xylocaine 2%, Adrenaline 1:200.00, AstraZeneca, Dilbeek, Belgium) was administered subcutaneously at skull level, the skin was disinfected with betadine solution, and the fur on the head was cut. We applied the ophthalmic gel (Vidisic gel 10 g, Bausch + Lomb Pharm SA, Bridgewater, NJ, USA) on the animals’ eyes to avoid drying during the procedure. A sterile scalpel was used to make a 0.5–1 cm incision in the middle of the scalp in the caudal direction. A Hamilton Syringe (701 RN, 30 gauge 1″ 10 µL) was cleaned with distilled water, 70% ethanol, and sterile PBS before and after each injection of cryptococci. The injection site was determined with respect to the bregma at 0.5 mm anterior and 2 mm right lateral. A 1 mm wide hole was drilled into the skull, and 1 µL, equal to 10^4^ cryptococci, was injected 3 mm deep from the dura using the Hamilton Syringe pump (QSI, Stoelting, Wood Dale, IL, USA) (settings: 0.5 µL/min; syringe ID: 700 HML 10 µL—0.485). After 10 min, the needle was withdrawn slowly and disinfected with 70% ethanol. The animal was placed on a heating pad, and the hole in the skull was filled with UV-activated dental filling resin (NanoFIl A2 3 g, AT&M Biomaterials, Beijing, China). The incision site was sutured and disinfected with betadine. Anesthesia was reversed using intraperitoneal injection of Atipamezol (0.1 to 1 mg/kg; Antisedan, Orion Pharma, Espoo, France). Animals recovered overnight in a warmed cage (half on a 37 °C heating pad) with wet food (Ssniff, Soest, Germany). After full recovery, the animals were reintroduced into the standard-of-practice housing. This murine model is based on previously described rat and murine models [41].

### 2.3. Morbidity Scoring

Post injection, the animals were evaluated every 1 to 3 days for neurotypical symptoms and weighed. The animal morbidity induced by the cryptococcal infection was scored using previously established criteria by Pool et al. [30]. The ethical endpoint was defined when 75% of the group reached an average level 2 in the morbidity scoring or 50% reached level 1 (Table 2). Additionally, euthanasia was performed if the lesion reached a volume greater than 75 mm^3^ or the lesion was not visible (0 mm^3^) on anatomical scans post-day 6 over two imaging time points defined as “full recovery”. In such cases, euthanasia criteria were met regardless of the morbidity scoring, although the scoring was still noted for that time point.

### 2.4. Magnetic Resonance Imaging

During all MRI scans, the mice were anesthetized using 1.5–2.0% isoflurane in 100% oxygen. After induction of anesthesia, the mice were positioned on a heated mouse bed while continuously receiving anesthesia [43]. We applied ophthalmic gel (Vidisic gel 10 g, Bausch + Lomb Pharm SA, USA) on the animal’s eyes to avoid drying during the scan. The animals’ breathing rate (60–120 breaths/min) and body temperature (37 ± 1 °C) were closely monitored (Small Animal Instruments, Stony Brook, NY, USA) and maintained at physiological levels. MRI scans were acquired every 2–4 days post-injection using a 9.4 T preclinical horizontal MRI scanner (Biospec 94/20, Bartlesville, OK, USA) with a linearly polarized ^1^H resonator for transmission and an actively decoupled mouse brain surface coil for reception (all Bruker Biospin, Ettlingen, Germany). The manufacturers’ software platform Paravision 360 v.3.0 was used to obtain all MRI scans. Localizer images were first acquired as an anatomical reference for the positioning of subsequent scans. Two-dimensional T_2_-weighted MRI scans were acquired using a rapid acquisition with relaxation enhancement (RARE) sequence. Parametric T_2_ relaxation maps were acquired with an MSME (Multi Slice Multi Echo) protocol and apparent diffusion coefficient (ADC) maps with a diffusion-weighted pulsed gradient spin echo (PSGE) sequence [38]. Parametric maps were acquired in the same orientation and with the same slice thickness as the axial T_2_-weighted MR images. In addition, ^1^H Point Resolved Spectroscopy (PRESS) magnetic resonance spectroscopy (MRS) data of a single volume of interest centered around the cryptococcal lesion were acquired from a predefined volume of 8 mm^3^; in addition, one animal per group was selected to acquire MR spectra in a predefined volume of 8 mm^3^ in the contralateral region [34,40]. The parameters used for MRI acquisition are listed in Table 3.

Parametric maps (T_2_ and ADC) were calculated using the Paravision 360 v.3.0 image sequence analysis tool. Scans were then converted into NIFTI format for further analysis.

### 2.5. Data Analysis MRI

The NIFTI-converted images were analyzed using ITKSnap v3.8 [44]. Lesion volume (mm^3^), T_2_ (ms), and ADC (mm^2^/s) values were determined by manually delineating the lesion over the entirety of its volume using the adaptive brush tool (brush size: 20—granularity: 0.15—smoothness: 0.30) on the whole anatomical scan and overlaying the segmentation on the co-registered parametric scans. A standardized circular region of interest of 0.56 mm^3^ was selected in the contralateral region of the brain, corresponding to the left striatum approximatively opposite to the center of the lesion, as a reference and quality control.

### 2.6. Data Analysis MRS

^1^H MR spectra were analyzed (Matlab, Statistics and Machine Learning Toolbox) with principal component analysis (PCA) in the region of interest [0.5–4.2] ppm, where common metabolites display their peaks, and region [5.0–5.5] ppm, where trehalose is present. With PCA, a set of uncorrelated variables (features) called principal components (PCs) are computed [44,45] so that most of the variance in the original dataset can be explained by the first PCs. Moreover, PCA provides insight into which metabolites/metabolic regions within the MR spectra are most relevant in describing the variance between the different groups.

### 2.7. Brain CFU Plating

When the endpoint was reached based on criteria determined by the morbidity scores, animals were sacrificed by intraperitoneal injection of a pentobarbital overdose (1 g/kg), and brains were extracted. The brain was weighed and homogenized in 600 µL sterile PBS. The homogenizer (Tissue Master 125 Watt W/7 mm probe, OMNI International, Kennesaw, GA, USA) was cleaned and disinfected between samples by activating it for 10 s in distilled water, 70% ethanol, and PBS. A 10-fold dilution series of the brain homogenates was made up to 1/10^6^. The samples were plated in triplicate on Sabouraud agar and incubated upside-down for 3–4 days at room temperature. The samples’ CFUs were manually counted, as described above. We later normalized the CFU through a logarithmic transformation of CFU divided by either brain weight (fungal burden) or lesion volume (lesion fungal density).

### 2.8. Statistical Analysis

Statistical analyses were performed by using Graphpad Prism v9.5.1. Log-rank (Mantel Cox) test was used to determine statistical differences in the endpoint curves. Mixed effects analysis with Geisser–Greenhouse correction for sphericity, followed by Dunnett’s multiple comparison tests, was used to determine longitudinal statistical differences in the morbidity scoring, lesion volume, T_2_, and ADC parameters. Correlations were assessed using linear regression and Pearson correlation coefficients.

## 3. Results

### 3.1. Cryptococcal Isolates of the Different Major Molecular Types Result in Variable Neurovirulence

Mice were stereotactically injected with one of the eight isolates of CN (VN I-IV) or CG (VG I-IV), each representing a different major molecular type. The morbidity scoring allowed the longitudinal assessment of neurological symptoms, highlighting differential progression and the manifestation of cryptococcoma in the brain. The neurotypical symptoms and morbidity induced by each isolate were variable between isolates (Figure 1A,B). Reaching the ethical endpoint was highly variable between molecular types, ranging between 5 and 20 days. Neurovirulence of the isolates was classified on the day of the average time needed for the morbidity score to reach level 2: “very high” (≤6 days), “high” (>6 days to ≤10 days), “medium” (>10 days to ≤15 days), “low” (≥16 days), or “non-neurovirulent” in case of full recovery (Figure 1C). We observed that CN isolates had medium to very high virulence, and CG isolates had either very high, low, or no virulence. Very high neurovirulent isolates showed generally a rapid decrease in the health score within 2–3 days after infection. On the contrary, isolates with lower neurovirulence showed variable trends with initial symptoms, such as reduced hygiene, being present only after 6–8 days post-infection. Non-neurovirulent isolates did not show any decay in the well-being of the animal. Additionally, normalized brain fungal burden calculated using the logarithm transformation of CFU divided by the brain weight at the endpoint (Figure 1D) was statistically different between CN and CG species (Appendix A).

### 3.2. Assessment of Cryptococcomas Using T_2_-Weighted Anatomical MRI

The development of cerebral cryptococcomas in this model is identified as hyperintensity and/or deformation of the right striatum in the mouse brain (Figure 2A,B). The shape, dimension, and evolution of lesion formation depended on the molecular type (Appendix A). Analysis of T_2_-weighted MRI showed that the lesion volume increases over time with comparable growth rates, resulting in lower virulent isolates manifesting larger lesion dimensions at the endpoint (Figure 2C,D). The differences in the lesion curves between the different molecular types were not statistically significant (Figure 2C,D).

Mixed effects statistical analysis indicated that lesion dimension is significantly related to the time post-injection rather than the neurovirulence scoring of the different isolates (Table 4), and VG III was excluded from the statistical analysis due to its non-neurovirulent property in this model.

When considering the endpoint, we observed a statistically relevant difference across isolates. Specifically, isolates with lower virulence tend to exhibit increased lesion volumes at the endpoint compared with those with higher virulence (Figure 3A,B). An additional biomarker of relative fungal burden is the image-based calculation of lesion density (LD) obtained using the logarithm transformation of CFU divided by the lesion volume at the endpoint (Figure 3C,D). When the cell densities of the major molecular types of CN in the lesions are compared with the cell densities of the major molecular types of CG, the analysis shows a higher fungal cell concentration in CN lesions (Figure 3E).

### 3.3. PCA Analysis of MRS Data at Endpoint

With PCA, we identify the metabolic regions of the acquired MR spectra that show variations that could be explored to discriminate between the different major molecular types of CN and CG. Due to the variable dimension of the lesion at the endpoint (Figure 3A,B), there was a variable partial volume effect, adding contribution from surrounding tissue to the lesion spectra that was ultimately identified as one of the PCs (Appendix A). We observed a better separation between CN and CG if the lesion volume is higher, and thus, the acquired MR spectra are a better representation of the fungal lesion formation. Consequently, there was a clear separation of VN I, being represented by higher lesion volume cases, with elevated peaks in the trehalose region. For VG III, due to its non-neurovirulent property in this model, the acquired MR spectra exhibited normal tissue metabolic features that group apart from all other cases.

### 3.4. Longitudinal Quantitative Parametric Evaluation of Cryptococcomas Show Molecular Type Dependent T_2_ Relaxometry and Disease Dependent Diffusivity

We tested the possibility of differentiating between different major molecular types of lesion-forming isolates of the CN/CG species complexes using qMRI. We investigated the longitudinal evolution of the fungal lesion in vivo (Figure 4A,B) and observed that T_2_ values were relatively consistent over time within each molecular type (Appendix A), indicating that the viscosity of the fungal lesion is approximatively consistent over time but different between different isolates/major molecular types (Figure 4C–E).

ADC values generally increase with the evolution of the disease (Figure 5A,B), showing higher values in CG species but similar values at the endpoint across the different major molecular types (Figure 5C–E). Compared with tissue measured in the contralateral hemisphere, water motility is higher in the cryptococcal lesions (Appendix A).

We further compared the qMRI parameters of the lesions at the endpoint, using the contralateral side of the brain as control. Both isolates of CN and CG show an increased T_2_ and water diffusivity (ADC) compared with the contralateral brain (Figure 6). We observed generally higher T_2_ values in the CG isolates compared with CN (Figure 6A), as well as statistically relevant differences between the major molecular types within each species complex (Figure 6B,C). We observed a similar trend of higher ADC values in the CG isolates compared with CN (Figure 6D), but on the contrary, we do not observe statistical differences between the molecular types of each species complex at the endpoint (Figure 6E,F). Both parametric values inversely correlate with lesion density, calculated using the logarithm transformation of CFU divided by lesion volume at the endpoint (Figure 6G,H). The different fungal density and molecular composition of the fungal lesion causes a change in the qMRI parameters distinguishable between CN and CG, with CN isolates having a more compact and viscous lesion environment compared with the looser structure of the CG isolates (Figure 6I,J).

## 4. Discussion

In this study, we provide a systematic in vivo evaluation of differences in neurovirulence induced by eight isolates representing the eight major molecular types of the CN/CG species complex. Many studies have addressed molecular and phenotypic/molecular differences between isolates of the CN/CG species complexes in culture, resulting in various proposals for their taxonomic classification [10,12,13,14,17,18,46,47,48,49]. To our knowledge, none of them have addressed differences in disease manifestation and pathogenesis in vivo. Contrary to the most commonly used laboratory reference strains (for CN strain H99 (VN I) and for CG strain R265 (VG II)), clinical and veterinarian isolates show differences in morbidity, and as such, we observed variable degrees of neurovirulence between the herein studied isolates representing the different major molecular types. We also confirmed that, on average, the CN species cause a higher fungal burden in the brain compared with the CG isolates at the endpoint using CFU counts. While these traditional evaluation methods of animal models, including morbidity scoring, survival curves, and CFU counts, provide useful information, in vivo imaging methods like MRI enable valuable insight into neurovirulence prior to endpoint. Specifically, MRI provides quantitative and multi-parametric information on clinically relevant parameters (lesion formation), cellular content (cell density, viscosity) and metabolism in a non-invasive manner at multiple time points for individual animals or patients [5,34,35,38]. The ability to delineate variable degrees of neurovirulence caused by the different isolates in a single focal lesion murine model from initial onset through endpoint is imperative for further understanding isolate-specific disease progression and kinetics. In addition, different isolates/major molecular types need to be studied before overall conclusions concerning the differentiation of the major molecular types by MRI methods can be concluded.

Lesion formation, including growth curves, lesion dimensions, and morphological differences in the manifestation of the focal fungal disease in the mouse brain, was achieved by using conventional anatomical MRI. Distinct patterns of lesion volume increase with comparable growth rates, as well as distinct growth patterns and hyperintensities for each of the isolates, were demonstrated. Trends in lesion density were comparable to the gold standard of CFU counts and showed differences between the isolates of the different major molecular types.

So-called advanced, quantitative MRI methods allow for the further characterization of lesions based on their physical properties. For example, T_2_ MR relaxometry provides a measure for the mobility of molecules like water and lipids due to the different degrees of water being bound to macromolecules and various degrees of viscosity [50]. Similarly, the determination of apparent diffusion coefficients (ADCs) by using diffusion MRI provides a measure for the random motion (diffusion) of water molecules in different tissues, thus highlighting pathology-derived alterations, namely differences in cell density and intercellular space [51,52]. While T_2_ and ADC values are frequently used for the characterization of tumor tissue, applications for fungal lesions remain sparse. In a previous study, we correlated these values with the composition of cryptococcomas, namely cryptococcal cell density and the fungal capsule size in the frequently used reference strains of *C. neoformans* and *C. gattii*, H99 and R265, respectively [38]. It is of interest to note that T_2_ and ADC values for H99 and R265 in this previous study were similar to the isolates of their respective molecular types (VN I and VG II) used in this study. This is the first indication that variability between isolates of the same molecular type might be less than between isolates of different molecular types. The MR properties of cryptococcomas correspond to their typical fluid-filled, pseudocystic lesion containing a variable amount of cryptococci. Our group previously compared the unique ADC and T_2_ values in focal cerebral cryptococcal lesions to in vivo fluorescence imaging of the two reference strains [38]. In this multi-modal study, we correlated cellular properties, like cell dimension and cell density, obtained with in vivo microscopy with qMRI findings. Higher cell densities and smaller capsule sizes were indicated for H99 when compared to R265. Our work here supports and generalizes these previous findings, as T_2_ and ADC values are significantly smaller for the molecular types of CN when compared to CG, indicating less intercellular space and higher cellular density. This previous study focused on a limited selection of fluorescent isolates, whereas here, we expand on the association between multiple clinical and veterinarian isolates, their disease manifestation, and image-based biomarkers. It is of interest to note that endpoint ADC values of the CN and CG isolates were comparable, even if the disease progression for each of these isolates was different. Exceptions are VG III and VG IV, where either no brain lesions were formed (VG III) or a very small lesion was formed (VG IV). The latter results in possibly biased ADC values due to partial volume effects resulting from the small size of the lesion relative to image slice thickness. Although more isolates per molecular type should be tested in the future, the work presented here supports the necessity to investigate not only microbiological and molecular tests for classifying isolates of the CN/CG species complexes but also experimental data on pathogenesis and physical properties of different isolates in in vivo models to gain complex characteristics to guide patient-specific treatment. Although no sham-injected control group was included in this study, previous work has shown that injection of PBS or non-proliferating cells results in only minor temporary changes in MRI (hypointense contrast due to needle track), which resolve within 3 to 4 days [39,53,54].

Further investigation of additional isolates also applied to different, more physiological models such as intravenous or intranasal administration would substantiate if this observation at a species and molecular type level can be generalized, adding fundamental data for species description. We chose the intracranial model in order to (1) minimize partial volume effects during the quantification of parametric qMRI data and (2) due to the availability of reference data from the H99 and R265 isolates in this model [34,38,39]. The relevance of partial volume effects became apparent for the quantification of metabolic data retrieved from the MRS experiments, where classification attempts were hampered by substantial contributions of surrounding brain tissue in the case of small lesions. The development of a quantitative method to overcome the substantial partial volume effects of measuring metabolism within the lesion in this model is underway. Regardless, PCA has shown to be a useful metric to identify the components of fungal metabolism and brain partial volume effects.

In conclusion, we demonstrate that clinically translatable qMRI techniques provide information on differences between CN and CG non-invasively, which are also based on differences in cell density in lesions and capsule size. While differences between molecular types were observed in terms of pathogenicity and neurological manifestation, more isolates per molecular type need to be studied. This foundational work can be easily translated to the clinic, which may add quantitative image-based parameters to optimize disease follow-up and, ideally, treatment thereof, as also demonstrated in other preclinical models [23,34,55]. Thereby, the qMRI approach provides non-invasive in vivo read-outs of variability, enabling insight into the biophysical properties caused by the different molecular types of the CN/CG species complexes.

## Figures and Tables

**Figure 1 jof-10-00593-f001:**
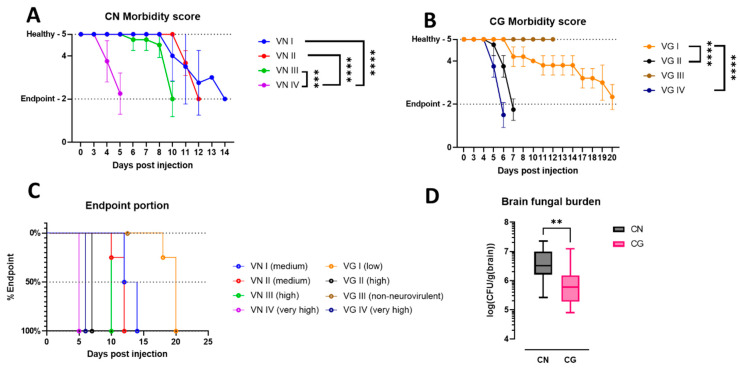
**Neurovirulence evaluation of BALB/cAnNCrl mice infected by stereotactic injection with cryptococcal isolates of the different major molecular types.** (**A**,**B**): The graphs show a longitudinal assessment of neurotypical symptoms in mice stereotactically infected with various CN (VN I–IV) or CG (VG I–IV) molecular types. Each graph shows the morbidity scoring per mouse group with standard deviation (*p*-values: *** < 0.001, **** < 0.0001). (**C**): The isolates were classified on the day of the average time needed for the morbidity score to reach a level of 2. “very high” (≤6 days), “high” (>6 days to ≤10 days), “medium” (>10 days to ≤15 days), “low” (≥16 days), or “non-neurovirulent” due to full recovery. (**D**): normalized brain fungal burden (Colony forming units divided by brain weight) shows a higher fungal burden in the CN species compared to the CG, Unpaired Welch’s *t* test (*p*-value = 0.0059 **).

**Figure 2 jof-10-00593-f002:**
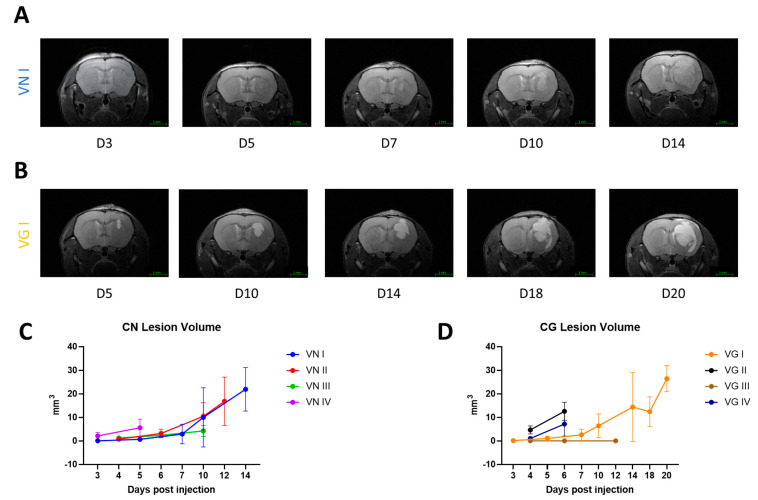
**Longitudinal T_2–_weighted MR images of two representative CN/CG isolates and assessment of cryptococcomas lesion volume induced by isolates of the different major molecular types.** (**A**). Longitudinal T_2_–weighted MR images in the axial orientation of a representative animal infected with the VN I isolate. (**B**). Longitudinal T_2_–weighted MR images in the axial orientation of a representative animal infected with the VG I isolate. The graphs in (**C**,**D**) show the longitudinal evaluation of the lesion volume induced by the different CN (VN I–IV) or CG (VG I–IV) molecular types. Each time point shows the average lesion volume with standard deviation (n = 4).

**Figure 3 jof-10-00593-f003:**
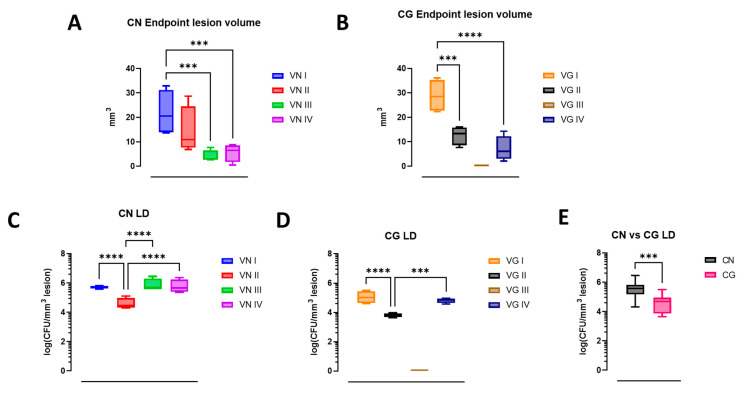
**Cryptococcal lesion volume and density at endpoint induced by different molecular types.** (**A**,**B**) boxplots show endpoint median lesion volume distribution, standard deviation and statistical differences of CN and CG isolates, respectively (*p*-values: *** < 0.001). The graphs (**C**,**D**) show lesion density (LD), calculated by the logarithm of CFU/lesion volume. One-way ANOVA with Fisher LSD test. (**E**) Comparison of lesion density (LD) by grouping all CN and all CG isolates. Unpaired Welch’s *t* test. (*p*-values: *** < 0.001, **** < 0.0001). VG III is shown for reference in graphs (**B**,**D**); however, since it is a non-neurovirulent isolate, it was not included in (**E**) or any statistical analysis.

**Figure 4 jof-10-00593-f004:**
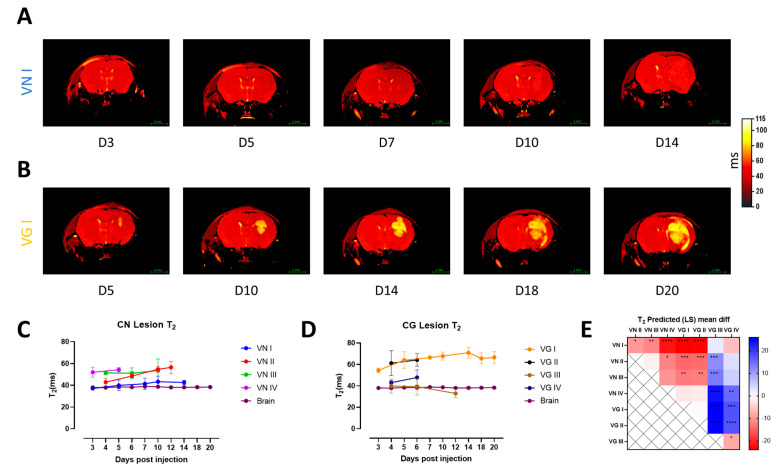
**Evaluation of T_2_ values in the lesion material and in the contralateral hemisphere in a single brain lesion model of cryptococcosis.** (**A**,**B**) Longitudinal T_2_ maps of lesions representing one CN and one CG–relevant molecular type. (**C**,**D**) Longitudinal assessment of T_2_ values in the lesion and in the contralateral brain as reference for isolates of CN and CG species complexes. (**E**) Correlation matrix of T_2_ values shows predicted mean difference between all isolates and relative statistical difference expressed in *p*–value in stars. Statistical analysis was performed with mixed effect analysis 2–way ANOVA with Tukey’s multiple comparison. (*p*–values: * < 0.05, ** < 0.01, *** < 0.001, **** < 0.0001).

**Figure 5 jof-10-00593-f005:**
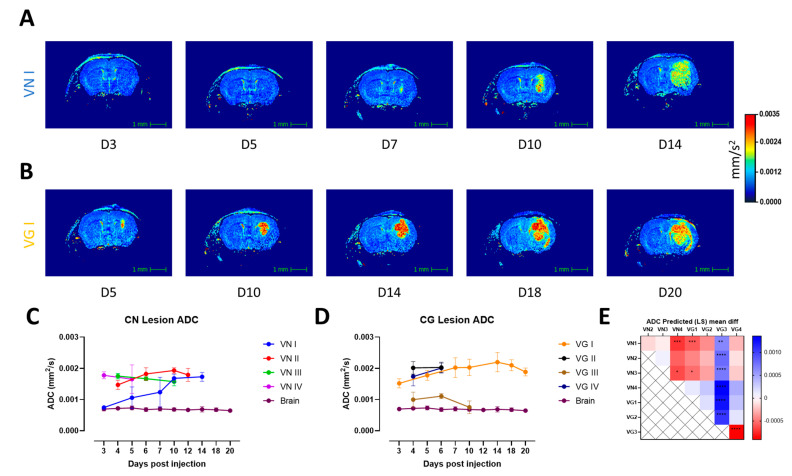
**Evaluation of ADC values in the lesion material and contralateral hemisphere in a single lesion brain model of cryptococcosis.** (**A**,**B**) Longitudinal ADC maps of lesions representing one CN and one CG (**B**) relevant molecular type. (**C**,**D**) Longitudinal assessment of ADC values in the lesion and in the contralateral brain for reference for isolates of CN and CG species complexes. (**E**) Correlation matrix of ADC values shows predicted mean difference between all isolates and relative statistical difference expressed in *p*–value in stars. Statistical analysis was performed with mixed effect analysis 2–way ANOVA with Tukey’s multiple comparison. (*p*–values: * < 0.05, ** < 0.01, *** < 0.001, **** < 0.0001).

**Figure 6 jof-10-00593-f006:**
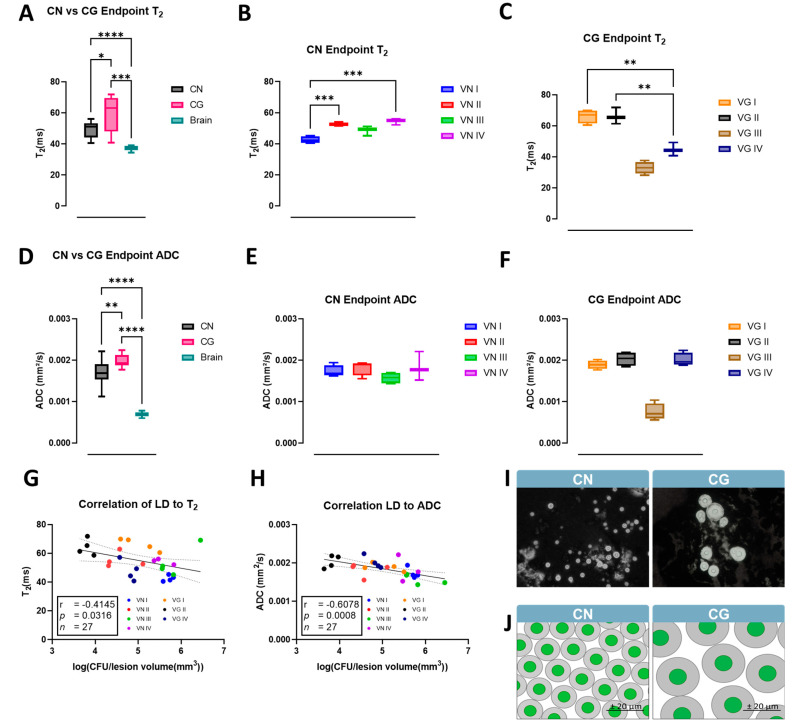
**qMRI analysis (T_2_ and ADC values) of the lesions at endpoint induced by isolates of different molecular types and correlation with lesion density (LD).** (**A**). Comparison between average CN, CG, and contralateral brain T_2_ values. (**B**,**C**). T_2_ values of CN and CG at endpoint show statistical differences between isolates. (**D**). Comparison between ADC values of CN, CG, and contralateral side. Statistical differences were calculated with one–way ANOVA with Welch’s multiple comparison test (*p*–values: * < 0.05, ** < 0.01, *** < 0.001, **** < 0.0001). VG III is shown for reference but is not included in the statistical analysis. (**E**,**F**). ADC values of CN and CG at endpoint with statistical differences between isolates. (**G**,**H**). Graphs show Pearson linear correlation (r) of T_2_ and ADC values to fungal lesion density (LD). (**I**). India ink stainings confirm a reduction in cell density within CG compared to CN, supporting the association between the increased capsule size and T_2_/ADC values. (**J**). Lesion cell structure model that links decreased cell density in CG compared to CN with increased capsule size and T_2_/ADC values as proposed by Vanherp et al. [38].

**Table 1 jof-10-00593-t001:** List of CN/CG molecular types used for in vivo murine experiments.

Species	Isolate	Molecular Type	Origin
CN	WM 148	VN I	Clinical
CN	WM 626	VN II	Clinical
CN	WM 628	VN III	Clinical
CN	WM 629	VN IV	Clinical
CG	WM 179	VG I	Clinical
CG	WM 04.71	VG II	Veterinarian
CG	WM 183	VG III	Clinical
CG	WM 779	VG IV	Veterinarian

**Table 2 jof-10-00593-t002:** Morbidity scoring and co-respective behavior. Adapted from Pool et al. [30].

Score	Condition	Related Behavior
5	Alert	Rearing, normal gait, normal exploration, nesting
4	Lethargy	Reduced grooming, but active/responsive/exploring
3	Reduced activity	Movement only when prodded, poor hygiene
2	Moribund	Hunchback, irregular or unstable gait, eye discharge together with photophobia, hydrocephaly
1	Morbid	Respiratory distress, inactive, immobile, unable to right self, >20% weight loss

**Table 3 jof-10-00593-t003:** MRI acquisition protocol and relative parameters.

	T_2_-Weighted Anatomical	T_2_-Weighted Anatomical	T_2_ Mapping	ADC Mapping	^1^H MRS
**Orientation**	Coronal	Axial	Axial	Axial	3D
**MR sequence**	RARE	RARE	MSME	PGSE	PRESS
**RARE factor**	8	8			
**TR/TE effective (ms)**	2500/42	3000/31.2	3000/…	1000/30	2000/20
**Number of slices**	9	24	24	24	1
**Slice thickness (mm)**	0.5	0.5	0.5	0.5	2
**FOV (cm)**	2 × 2	2 × 2	2 × 2	2 × 2	0.2 × 0.2 × 0.2
**In-plane resolution (mm^2^)**	200	200	200	200	/
**Acquisition time**	45 s	2 min 30 s	6 min 40 s	8 min 56 s	8 min 54 s
**Sequence-specific details**			10 TEs from 10 to 100 ms with 10 ms increments	3 b-values: 150, 500 and 1250 s/mm^2^	256 averages, VAPOR water-suppression

**Table 4 jof-10-00593-t004:** Mixed effects statistical analysis of lesion volume comparison with Geisser–Greenhouse correction.

Fixed Effects (Type III)	*p* Value	*p* Value Summary	Statistically Significant (*p* < 0.05)?	F (DFn, DFd)
**Time**	<0.0001	****	Yes	F (9, 60) = 19.30
**Molecular type**	0.1959	ns	No	F (6, 21) = 1.602

## Data Availability

All the data is displayed in this paper. Raw data is available on request, please contact uwe.himmelreich@kuleuven.be.

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
