# Peer review of "Quantitative MRI of a Cerebral Cryptococcoma Mouse Model for In Vivo Distinction between Different Cryptococcal Molecular Types"

_jof, 2024, doi:10.3390/jof10080593_

Round 1

Reviewer 1 Report

In this manuscript, authors evaluated clinically translatable quantitative magnetic resonance imaging techniques that provide information on differences between CN and CG non-invasively. This manuscript contains noteworthy insight on the pathogenesis, lesion formation and biophysical properties caused by the different molecular types of the CN/CG species complexes in the mouse brain. Nevertheless, some comments are suggested to be considered:

1.  In the "Materials and Methods-Strains and Cell Line" section, the study included a total of 8 CN/CG molecular types strains. Please provide more details on the characteristics of these strains, such as Capsule formation and thickness analysis, Melanin and urease production assay, and incubations conditions.

2. In the "Materials and Methods- Stereotactic injection in mice" section, female BALB/cAnNCrl mice (n=4 per isolate, 32 mice total) were used for all experiments. However, there is no blank control group?

3. In the " Results- Assessment of cryptococcomas using T2-weighted anatomical MRI " section, The development of cerebral cryptococcomas in this model are identified as hyper-intensity and/or deformation of the right striatum in the mouse brain (Figure 2 A,B). How about the characteristics of other types, such as VN II, VNIII or VGII? The same issues are showing in the Figure 4 and Figure 5.

In this manuscript, authors evaluated clinically translatable quantitative magnetic resonance imaging techniques that provide information on differences between CN and CG non-invasively. This manuscript contains noteworthy insight on the pathogenesis, lesion formation and biophysical properties caused by the different molecular types of the CN/CG species complexes in the mouse brain. Nevertheless, some comments are suggested to be considered:

1.  In the "Materials and Methods-Strains and Cell Line" section, the study included a total of 8 CN/CG molecular types strains. Please provide more details on the characteristics of these strains, such as Capsule formation and thickness analysis, Melanin and urease production assay, and incubations conditions.

2. In the "Materials and Methods- Stereotactic injection in mice" section, female BALB/cAnNCrl mice (n=4 per isolate, 32 mice total) were used for all experiments. However, there is no blank control group?

3. In the " Results- Assessment of cryptococcomas using T2-weighted anatomical MRI " section, The development of cerebral cryptococcomas in this model are identified as hyper-intensity and/or deformation of the right striatum in the mouse brain (Figure 2 A,B). How about the characteristics of other types, such as VN II, VNIII or VGII? The same issues are showing in the Figure 4 and Figure 5.

Author Response

(1) In the "Materials and Methods-Strains and Cell Line" section, the study included a total of 8 CN/CG molecular types strains. Please provide more details on the characteristics of these strains, such as Capsule formation and thickness analysis, Melanin and urease production assay, and incubations conditions.

Response: We thank the reviewer for this suggestion. While all isolates have been characterized in previous publications, we understand that in particular further information on the capsule and cell size are related to our manuscript. Therefore, we have performed additional experiments to also include such data. We have added a more detailed supplementary table in the supplementary material, which includes cell and capsule dimension in culture as well as references to previous publications (see Supplementary Table S1).

(2) In the "Materials and Methods- Stereotactic injection in mice" section, female BALB/cAnNCrl mice (n=4 per isolate, 32 mice total) were used for all experiments. However, there is no blank control group?

Response: While the main purpose of our study was the comparison of lesions caused by the different isolates and not to compare cryptococcoma to other conditions, we recognize the significance of including control/ sham injection groups. At the same time, we are bound to legal regulations concerning the use of experimental animal models. In this regards, we were asked by the responsible ethics committee to make use of previously performed experiments. For previous experiments on brain tumour models, stem cell implantation and focal brain infections, sham groups (PBS injection in the brain using similar coordinates as in the current experiment) were used. The analysis of the respective MRI data indicated the presence of a few pixel-wise hypointensities resulting from the injection procedure (needle track), which completely clear up three days post-injection. In the revised version of the manuscript, we have referred to these previously performed control experiments (line 443-446).

(3) In the " Results- Assessment of cryptococcomas using T2-weighted anatomical MRI " section, The development of cerebral cryptococcomas in this model are identified as hyper-intensity and/or deformation of the right striatum in the mouse brain (Figure 2 A,B). How about the characteristics of other types, such as VN II, VNIII or VGII? The same issues are showing in the Figure 4 and Figure 5.

Response: Thank you for pointing this out. Examples for the parametric maps (T2 and ADC) of all molecular types were already included in the supplementary figures of the originally submitted manuscript. In response to the reviewer’s comment, we have also added anatomical images to these supplementary figures.

Reviewer 2 Report

This is a solid piece of research that fits well in this journal. I enjoyed the reading and most importantly, it addresses an important gap in our knowledge of Cryptococcus and cryptococcosis. The methodology contains enough information to allow reproducibility, the results are plausible and the conclusions align with the results. In my more than 20 years of career, I haven't found a manuscript with no observations to make. I am glad that I have finally found it.

This is a solid piece of research that fits well in this journal. I enjoyed the reading and most importantly, it addresses an important gap in our knowledge of Cryptococcus and cryptococcosis. The methodology contains enough information to allow reproducibility, the results are plausible and the conclusions align with the results. In my more than 20 years of career, I haven't found a manuscript with no observations to make. I am glad that I have finally found it.

Author Response

This is a solid piece of research that fits well in this journal. I enjoyed the reading and most importantly, it addresses an important gap in our knowledge of Cryptococcus and cryptococcosis. The methodology contains enough information to allow reproducibility, the results are plausible and the conclusions align with the results. In my more than 20 years of career, I haven't found a manuscript with no observations to make. I am glad that I have finally found it.

Response: Thank you for your kind and encouraging feedback. We are thrilled to hear that you found our research valuable and well-executed.

Reviewer 3 Report

Overall, the research work is carried out with scientific rigor.

The approach and techniques used to evaluate a fungal disease are noteworthy, as they could be extrapolated to the disease in humans.

I was unable to access the supplementary material.

Line 41: Define WHO

Section methods: change RPM --> x g

lines 127-144: References of this method.

lines 161-164. Reference

lines 165-180. Reference

Fig 6. I suggest placing the CN and CG microscopy images. Could you set some representative originals and then, you can place the cartoons where the dimensions of the capsule are represented?

I suggest, if possible, mentioning some features of the environmental isolate VGIII, which in the analyses performed does not behave similarly to clinical isolates.

Author Response

(1) I was unable to access the supplementary material.

Line 41: Define WHO

Section methods: change RPM --> x g

lines 127-144: References of this method.

lines 161-164. Reference

lines 165-180. Reference

Response: Thank you for your thorough review and positive feedback.

We apologize that the supplementary material was not accessible. We will ensure that the supplementary material is accessible for the revised manuscript.

We applied all suggested additions/ changes to the revised manuscript:

Definition of World Health Organization in line 41.

Changed RPM to x g in the methods section. Lines 106-107

Added references on lines 127-144, 161-164, and 165-180. Now lines 145, 163, 176 and 181

(2) Fig 6. I suggest placing the CN and CG microscopy images. Could you set some representative originals and then, you can place the cartoons where the dimensions of the capsule are represented?

Response: Thank you for your valuable suggestion. We selected examples for two representative molecular types and performed India ink stainings. These images have been added to Figure 6 of the revised manuscript.

(3) I suggest, if possible, mentioning some features of the environmental isolate VGIII, which in the analyses performed does not behave similarly to clinical isolates.

Response: Thank you for your suggestion. During our in vitro scans, we have observed that VG III has a smaller capsule thickness and a smaller cell dimension compared to the other isolates. This information along with some other information and reference to previously published data has been added to the supplementary Table S1 (see also response to reviewer 1, line 264-266; 320-321 and 336-337).